# The Impact of Non-optimum Ambient Temperature on Years of Life Lost: A Multi-county Observational Study in Hunan, China

**DOI:** 10.3390/ijerph17082699

**Published:** 2020-04-14

**Authors:** Ling-Shuang Lv, Dong-Hui Jin, Wen-Jun Ma, Tao Liu, Yi-Qing Xu, Xing-E Zhang, Chun-Liang Zhou

**Affiliations:** 1Hunan Provincial Center for Disease Control and Prevention, Changsha 410005, China; lingshuang_lv@163.com (L.-S.L.); hncdcmbkjdh@163.com (D.-H.J.); xu0510yiqing@163.com (Y.-Q.X.); yesgoshopping@163.com (X.-E.Z.); 2Guangdong Provincial Institute of Public Health, Guangdong Provincial Center for Disease Control and Prevention, Guangzhou 511430, China; mawj@gdiph.org.cn (W.-J.M.); gztt_2002@163.com (T.L.)

**Keywords:** Ambient temperature, year of life lost, ANUSPLIN, attributable fraction

## Abstract

The ambient temperature–health relationship is of growing interest as the climate changes. Previous studies have examined the association between ambient temperature and mortality or morbidity, however, there is little literature available on the ambient temperature effects on year of life lost (YLL). Thus, we aimed to quantify the YLL attributable to non-optimum ambient temperature. We obtained data from 1 January 2013 to 31 December 2017 of 70 counties in Hunan, China. In order to combine the effects of each county, we used YLL rate as a health outcome indicator. The YLL rate was equal to the total YLL divided by the population of each county, and multiplied by 100,000. We estimated the associations between ambient temperature and YLL with a distributed lag non-linear model (DNLM) in a single county, and then pooled them in a multivariate meta-regression. The daily mean YLL rates were 22.62 y/(p·100,000), 10.14 y/(p·100,000) and 2.33 y/(p·100,000) within the study period for non-accidental, cardiovascular, and respiratory disease death. Ambient temperature was responsible for advancing a substantial fraction of YLL, with attributable fractions of 10.73% (4.36–17.09%) and 16.44% (9.09–23.79%) for non-accidental and cardiovascular disease death, respectively. However, the ambient temperature effect was not significantly for respiratory disease death, corresponding to 5.47% (−2.65–13.60%). Most of the YLL burden was caused by a cold temperature than the optimum temperature, with an overall estimate of 10.27% (4.52–16.03%) and 15.94% (8.82–23.05%) for non-accidental and cardiovascular disease death, respectively. Cold and heat temperature-related YLLs were higher in the elderly and females than the young and males. Extreme cold temperature had an effect on all age groups in different kinds of disease-caused death. This study highlights that general preventative measures could be important for moderate temperatures, whereas quick and effective measures should be provided for extreme temperatures.

## 1. Introduction

The fifth report of the Intergovernmental Panel on Climate Change (IPCC) stated that over the period 1880 to 2012, the global temperature showed a warming of 0.85 °C (0.65 °C to 1.06 °C) [1]. The temperature–health relationship is of growing interest after episodes of extreme weather [2]. Many epidemiological studies have provided evidence for the association between ambient temperature and health outcomes [3,4].

At present, research on the association between ambient temperature and human health has mainly focused on the effects of extreme heat weather, and public health plans have implemented policies and interventions designed almost exclusively for heat wave [5,6]. However, a cold spell risk is generally ignored compared with previous studies focusing on heat waves [7,8]. Some studies have shown that the risk of extreme cold temperature was consistently higher than the risk of heat temperature on human health. Chen et al. [9] reported that 1.14% and 0.63% of non-accidental total mortality was attributable to extreme cold and extreme heat temperatures, respectively. In UK regions, cold-related mortality accounts for more than one order of magnitude than heat-related mortality. Every year, approximately 33 and two deaths per 100,000 population in Australian cities are associated with cold and heat temperature, respectively [10]. Moreover, some studies have reported that the effect of days of extreme temperature was substantially less than that attributable to moderate temperature. A multi-country observational study demonstrated that 7.71% of mortality was attributable to non-optimum temperature in 384 locations within the study period between 1985 and 2012, where the extreme cold and heat temperatures were responsible for 0.86% of total mortality [11]. A previous study showed that a moderately cold temperature was associated with a higher attributable risk (6.3%, 1.1 to 11.1) than extremely cold temperatures, moderately heat temperatures, and extremely heat temperatures, each of which were less than 0.6% [12]. One study even suggested that the preventive efforts were the same for warm and cold days and heat and cold waves [13]. Thus, more studies should be extended to take account of the whole range of effects associated with ambient temperature. Some studies have been published concerning the impact of daily ambient temperature on mortality or morbidity [14,15]. However, those approaches failed to consider the differences in age, resulting in incomplete analysis of information. One approach to avoid the drawback is to consider year of life lost (YLL) as the outcome indicator instead of mortality. YLL, an important part of disability adjusted life years, is a measure of disease burden that uses the life expectancy. Compared with the traditional measure of mortality, YLL gives more weight to deaths among younger people [16]. YLL has been regarded as a more novel and precise indicator to evaluate the burden of disease and is used to identify and prioritize causes of premature deaths [17,18]. To date, the ambient temperature-related YLL has only been studied by a few investigators, and the effects among different cause-specific death and socioeconomic status are also less clear.

In this study, we aimed to quantify the YLL burden attributable to non-optimum ambient temperatures among different cause-specific death and socioeconomic status based on valid data from 2013 to 2017 of 70 counties in Hunan, China. We also investigated the effect of moderate and extreme temperatures on YLL.

## 2. Materials and Methods

### 2.1. Study Sites

Hunan province is located in central China, with an area of 211.8 thousand square kilometers and a population of 68.99 million in 2018. We collected daily all-cause death data for 70 counties (Figure 1) in Hunan, China, from 2013 to 2017. The population of the cities ranged from 24.71 thousand to 2.09 million.

### 2.2. Death Data Collection

We acquired daily counts of disease death from The National Disease Surveillance Points (DSPs) System of China from 1 January 2013 to 31 December, 2017. Causes of death were coded based on the tenth version of the International Classification of Diseases (ICD-10). We acquired daily counts of non-accidental (A00-R99), cardiovascular (I00-I99), and respiratory (J00-J99) disease mortality. Information on gender, birth date, date of death, and cause of death were provided.

### 2.3. Meteorological Interpolation Method

The interpolation software for meteorology data, ANUSPLIN (Fenner School of Environment and Society, Canberra, Australia), was used to interpolate the climate factor. ANUSPLIN is a tool for spatial interpolation of multiple climate data based on thin-plate smoothing spline interpolation theory [19]. This study attempts to use ANUSPLIN to interpolate average temperature, with source data from 698 meteorological stations in China. The raster data of average daily temperature was developed at resolution of 0.01 × 0.01. Latitude and longitude were treated as independent variables, and elevation was considered a covariate. The function was as follows:(1)$$Temp_{i}=f\left(lat_{i},lon_{i}\right)+b*alt_{i}+e_{i},$$
where *Temp_i_* is the average daily temperature at meteorological station; *f* () is function of the thin-plate spline; *lat_i_*, *lon_i_*, and *alt_i_* are latitude, longitude, and elevation for the meteorological station; *b* is the regression coefficient; and *e_i_* is the error. The interpolation method was verified by a 10-fold cross-over method.

Then, the daily average temperature of the 70 study locations was extracted from raster data. The same method was used to obtain the daily relative humidity.

### 2.4. YLL Calculation

The life table of Hunan province was calculated according to the method established by Qinglang Jiang [20], using the results of the latest China census 2010 survey. The life table of Hunan province is presented in Appendix A. We calculated YLL by matching age and sex to the life table for each death. The daily YLL was calculated by summing the YLL for all death that occurred on the same day. We stratified the daily YLL by death causes, gender, and age (0–64, 65+ years). The value of YLL for each research location was related to the local population. In order to combine the effects of each district and county, we used YLL rate as health outcome indicator. The YLL rate is equal to the total YLL divided by the population of each district, and multiplied by 100,000. The unit of YLL rate is y/(p·100,000).

### 2.5. Statistical Analysis

We used a two-stage approach to investigate the association between ambient temperature and YLL in the present study.

#### Stage-1: Quantify the general effect of daily ambient temperature

Distributed lag non-linear model (DLNM) was proposed [21], which is flexible enough to examine both non-linear and delayed effects simultaneously, after accounting for the strong collinearity of exposure variables. We applied a Gaussian regression model with DLNM to assess the non-linear exposure–response relationship and delayed effect of ambient temperature on YLL. The function is as following:(2)$$YLLR_{t}=\alpha +\beta cb+NS\;\left(time,\;df\;*\;T\right)+NS\;\left(rh,\;df\right)+\gamma Dow,$$
where *t* is the day of observation; *YLLR_t_* is the YLL rate at day *t*; *α* refers to the intercept; *β* is coefficients for *cb*; *cb* refers to a two-dimensional natural spline for daily mean temperature with a lagged 21 days, considering the effect of non-optimum temperature on death and harvesting effect [22]; *NS* represents natural cubic spline; *NS (time)* is used to control long-term trend and seasonality; df is determined by Akaike’s Information Criterion (AIC); T is the time span; *NS (rh)* is used to control the confounding effects of humidity; *γ* is the coefficients for *Dow*; and *Dow* refers to the day of the week. We modelled the exposure–response relationship with three internal knots placed at the 10th, 75th, and 90th percentile of county-specific temperature distributions.

#### Stage-2: Estimated overall cumulative exposure-response association

A meta-analysis based on random-effect model was used to pool the estimated association between ambient temperature and YLL at the provincial level. In order to quantitatively estimate the overall effect of ambient temperature on YLL in the 70 counties, we calculated the attributable fraction. The total attributable YLL rate caused by non-optimum temperature is given by the sum of the contributions from all the days of the series, and its ratio with the total YLL rate provides the total attributable fraction. The components attributable to cold and heat temperatures were computed by summing the subsets corresponding to days with temperature below or above the minimum mortality temperature (MMT), respectively. MMT is derived by the lowest point of the overall cumulative exposure–response curve, and it is interpreted as the optimal temperature characterized by the lowest risk of YLL rate. We also explored the YLL burden attributable to mild and extreme temperatures. Extreme heat and cold temperature were defined as the temperature above the 97.5th percentile and below the 2.5th percentile mean temperature. Moderate heat and cold temperature were defined as the temperature between the mean temperature and extreme heat and cold temperature.

### 2.6. Sensitivity Analysis

To check the sensitivity of analysis results to model parameters and evaluate the stability of the model, sensitivity analysis was conducted by changing the degree of freedom (df) for time (6–8 per year), humidity (3–5), and the maximum lag days for mean temperature (14 and 21 days). We deemed the model with the lowest AIC as the final model.

We used R software (University of Auckland, Auckland, New Zealand) version 3.6.2 to conduct data analysis. The relationship between temperature and YLL was estimated by using package “dlnm”. Multivariate meta-analysis was conducted by using package “mvmeta”.

## 3. Results

### 3.1. Characteristics for YLL and Meteorological Variables

Table 1 shows the summary statistics of daily YLL rate and weather conditions in different sub-populations from 2013 to 2017. The daily mean temperature was 17.74 °C, ranging from −3.52 °C to 34.49 °C. The daily relative humidity ranged from 30.72% to 100%, with a mean value of 77.78%. The daily mean values of daily YLL rate were 22.62 y/(p·100,000), 10.14 y/(p·100,000), and 2.33 y/(p·100,000) for non-accidental, cardiovascular, and respiratory diseases death.

### 3.2. Association of Temperature and YLL

Figure 2 shows the exposure–response curve between daily mean temperature and YLL rate. All the pooled curves were U-shaped, indicating that the non-optimum temperature was associated with increased YLL rate. The temperatures of 25.6 °C, 28.4 °C, and 23.1 °C were related to the minimum YLL rate for non-accidental, cardiovascular, and respiratory disease death, respectively. The association between ambient temperature and YLL rate was observed among different sub-populations. The YLL rates were found to be relatively higher among males and the elderly than females and people less than 65 years old in the three kinds of disease-specific death (Figure 3 and Figure 4).

### 3.3. The Attributable Risk of Ambient Temperature on YLL

When we calculated the attributable risk on YLL in different disease-specific populations, we found that the effects of ambient temperature were significantly associated with the increase of YLL for non-accidental and cardiovascular disease death, with an attributable fraction of 10.73% (4.36–17.09%) and 16.44% (9.09–23.79%), respectively. However, the effect was not significantly in respiratory disease death, with an attributable fraction of 5.47% (−2.65–13.60%). Cold temperature was responsible for most of the YLL for disease-specific death, with overall estimate of 10.27% (4.52–16.03%) and 15.94% (8.82–23.05%) for non-accidental and cardiovascular death, respectively. However, the risk attributable to heat temperature was quite low, with an overall estimate of 0.45% (−0.16–1.06%) and 0.50% (0.26–0.73%) for non-accidental and cardiovascular death, respectively. The attributable risk can be separated into components related to moderate and extreme temperature. Most of the YLL attributable to ambient temperature were related to moderate cold, with overall estimate of 9.26% (3.87–14.64%) and 14.55% (7.83–21.27%) for non-accidental and cardiovascular death, respectively. Extreme temperature was responsible for only a small fraction. We also calculated the attributable risk associated with temperature among different subgroups stratified by gender and age. The attributable fraction varied substantially among different subgroups, with the highest attributable fraction in female and the elderly aged 65+ years. Extreme cold temperature was significantly associated with YLL increase for all subgroups (Table 2).

### 3.4. Sensitivity Analysis

Sensitivity analysis was performed to check whether the results were robust to the specification of parameters in the model. Appendix A revealed that the AICs of the model for non-accidental disease death were stable when using different model values. When the df was seven for temperature, maximum lag day was 21, and the df was three for relative humidity, the model had the lowest AIC value of 13,283.90.

## 4. Discussion

This study presented the association between ambient temperature and YLL at 70 sites in Hunan, China. Its exposure–response curves were basically a U-shaped, which is consistent with previous studies with U or J-shaped relationships [23,24]. Both the increase of the high temperature and decrease of the low temperature increases the YLL for non-accidental, cardiovascular, and respiratory disease death.

Our findings showed that cold temperature had significant impact on YLL due to non-accidental and cardiovascular diseases, with the attributable fraction of 10.27% and 15.94%, respectively. The results indicated that 10.27% and 15.94% YLL of non-accidental and cardiovascular diseases in non-optimum temperature days could be attributed to non-optimum temperature. Heat temperature had a significant impact on YLL due to cardiovascular disease death, with an attributable fraction of 0.50%. However, the effects of cold and heat temperature were not significantly in respiratory disease death. The attributable risk for cardiovascular disease death in our study was different with other studies. One study showed that attributable fraction to cold effect was from 2.67 to 8.55, and the attributable fraction to heat effect was from 0.16 to 2.29 due to cardiovascular disease in China [25]. Li et al. [26] reported that mild cold temperature contributed the largest fraction to YLL (16.31%). Various underlying mechanisms have been postulated to explain the increased YLL risk associated with high and low ambient temperature. Cardiovascular mortality has been shown to increase in the winter. Exposure to cold temperatures was associated with cardiovascular death by inducing increases in systolic and diastolic blood pressures, platelet count and serum low density lipoprotein-cholesterol concentration, and initiating a mild inflammatory reaction [27,28]. In addition, the main underlying mechanism for heat temperature related YLL is that stress on the cardiovascular systems increases during periods of high ambient temperature, especially among elderly people with limited adaptive responses. In elderly people, the ability to thermoregulate body temperatures is reduced [29].

Most of the YLL burden was caused by colder temperatures than warmer temperatures. This difference was mainly caused by the high minimum-mortality percentile, with most of the mean daily temperatures lower than the optimum value. Lately, one study has also found that the lower temperatures could lead to an increase of the cumulative number of COVID-19 cases, whereas, higher temperatures could lead to a decrease of the cumulative number of cases [30]. Thus, from the public health perspective, the government should pay more attention to cold weather to reduce life lost. Our finding showed that moderate cold temperatures and extreme cold temperatures had a significant impact on YLL due to non-accidental disease, with the attributable fraction of 9.26% and 1.02%, respectively. The extreme heat temperatures had a significant impact on YLL due to non-accidental disease, with the attributable fraction of 0.23%. However, the moderate heat temperatures had no significant impact on YLL due to non-accidental disease. Overall, the most of YLL were caused by exposure to moderately heat and cold temperatures, and the contribution of extreme days was comparatively low. Our study suggested that public-health policies and adaptation measures should be extended and refocused to take into account of the whole range of effects associated with ambient temperature.

In the study, we calculated the attributable risk associated with temperature among different subgroups stratified by different demographic characteristics. The analysis of data from 70 locations provides evidence for ambient temperature related YLL in a wide range of populations with different genders and ages. Our results demonstrated that cold and heat temperatures had significant impacts on YLL for people aged up to 65 years. Moreover, people aged over 65 years were at higher YLL risk of both heat and cold temperatures than young group. Studies have shown that elderly people are the most vulnerable group to cold or heat temperatures [31,32]. This may be caused by relatively poorer physiological adaptation and pre-existing chronic diseases among the elderly. We found that extreme cold temperatures had effect on all age groups due to different disease-cause mortality. With the increasing societal resources on preventive cares for the elderly, the protecting measures for the youth to combat the harmful effects of extreme cold temperatures should not be ignored. Our results also indicated that both cold and heat temperature-associated YLLs were higher in females than males. Differences in socioeconomic factors and ability to withstand cold and heat stress between genders, may lead to the increase of sensitivity of women [33]. These findings provide evidence on temperature-related YLL between different sociodemographic characteristics, which have significant meaning for environmental and public health policy making.

In our study, some limitations must be acknowledged. First, this investigation only included the locations in Hunan province, China, and the assessment was mainly restricted to local area. Secondly, we used ANUSPLIN to interpolate the climate factor and the data become more accuracy. There could be measurement error, due to missing data of the use of air conditioning and the effect of urban heat island. Furthermore, we did not control for the impact of air pollution. Fourthly, the data covers only five years. In the future, we will collect data with longer time and more regions for the research.

## 5. Conclusions

Based on the data of 70 counties in Hunan, China, our study provides evidence that non-optimum temperature could increase the YLL. The temperature–YLL relationship varied in different disease-cause mortality and sociodemographic factors. Moreover, extreme cold temperatures had an effect on all age groups in different disease-cause deaths. This study suggests that public health interventions are necessary to minimize the health consequences of non-optimum temperatures. General preventative measures, such as planting trees and installing air conditioners, should be done to minimize the health consequences of moderate temperatures. Whereas quick and effective measures should be provided for extreme temperatures, such as starting the emergency system and protecting outdoor workers.

## Figures and Tables

**Figure 1 ijerph-17-02699-f001:**
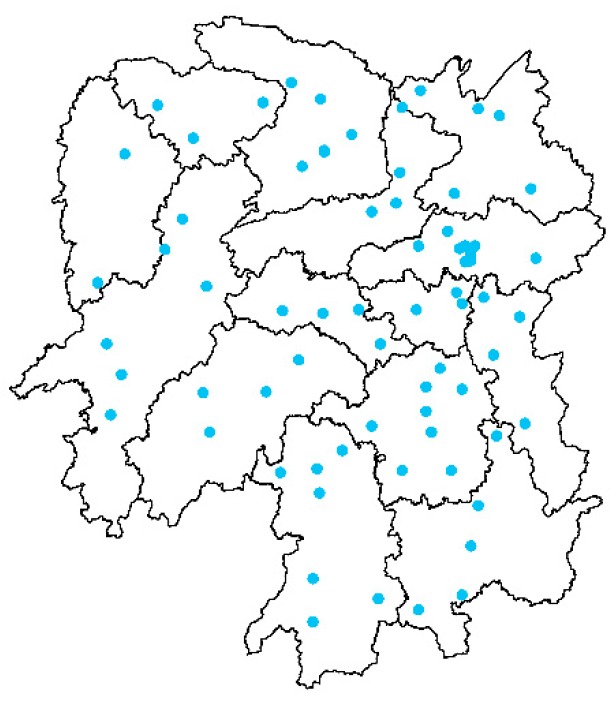
Distribution of 70 counties included in the present study. Blue dots represent the location of study counties.

**Figure 2 ijerph-17-02699-f002:**
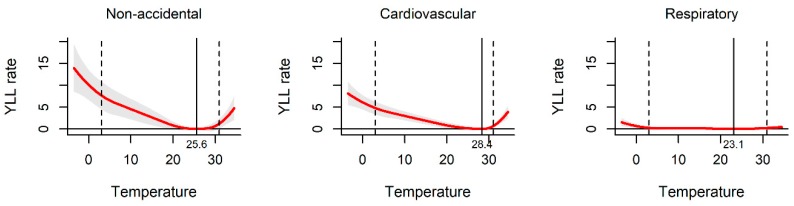
The effect of ambient temperature on year of life lost (YLL) in Hunan, China, 2013–2017. The solid red line shows the mean YLL rate, and the grey area shows the 95% confidence intervals. The solid black line represents minimum YLL rate temperature and the dashed black lines represents the 2.5th and 97.5th percentiles.

**Figure 3 ijerph-17-02699-f003:**
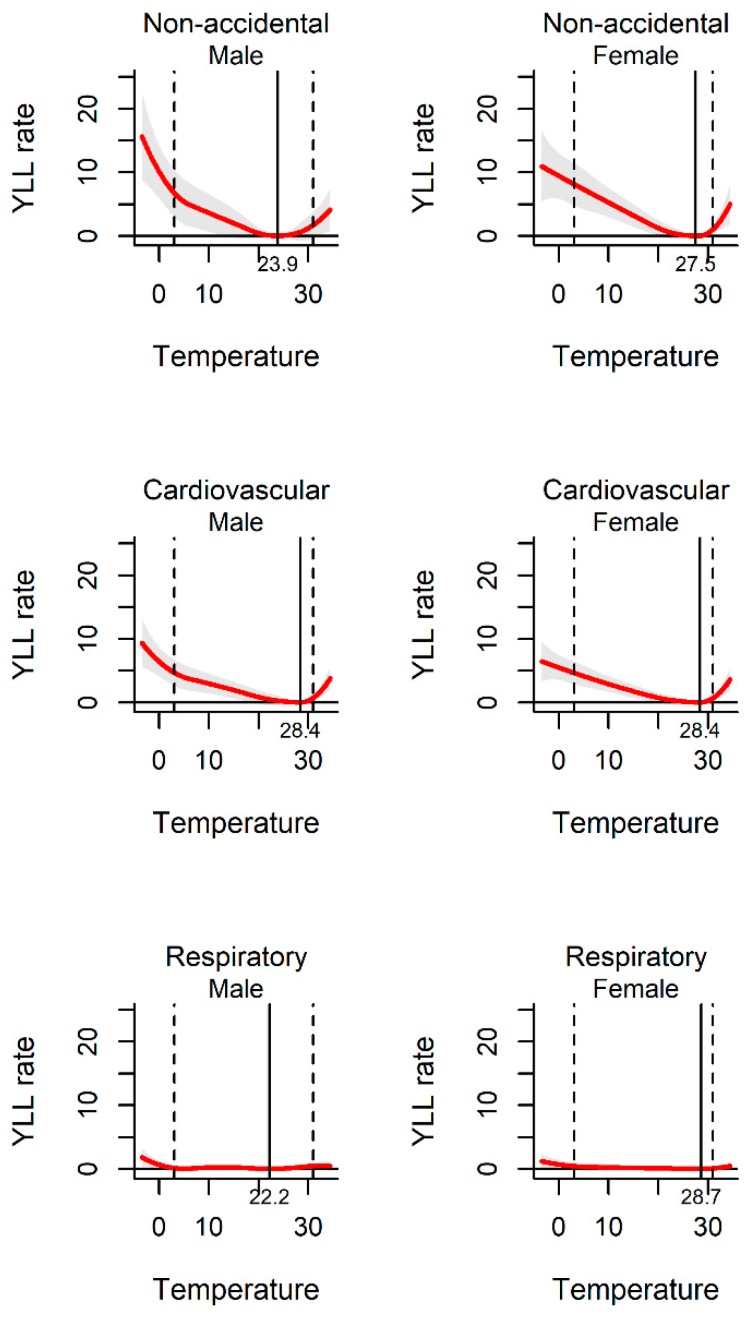
The association between ambient temperature and YLL rate separated by gender. The solid red line shows the mean YLL rate, and the grey area shows the 95% confidence intervals. The solid black line represents minimum YLL rate temperature and the dashed black lines represents the 2.5th and 97.5th percentiles.

**Figure 4 ijerph-17-02699-f004:**
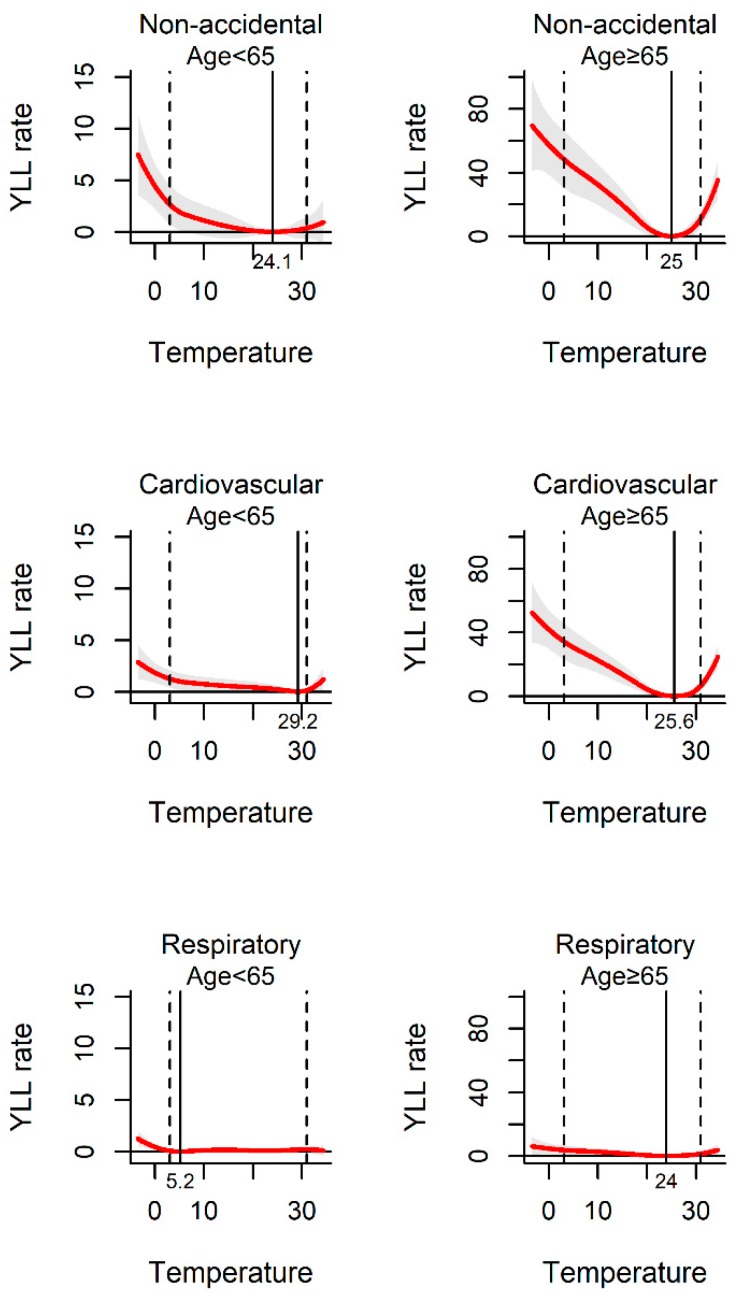
The association between ambient temperature and YLL rate separated by different age-group. The solid red line shows the mean YLL rate, and the grey area shows the 95% confidence intervals. The solid black line represents minimum YLL rate temperature and the dashed black lines represents the 2.5th and 97.5th percentiles.

**Table 1 ijerph-17-02699-t001:** Descriptive statistics of daily weather and year of life lost (YLL) rate from 2013 to 2017 in Hunan, China.

Characteristics	Min	P_2.5_	Median	Mean	P_97.5_	Max	SD
Meteorological							
Temperature (°C)	−3.52	10.41	18.55	17.74	24.74	34.49	8.30
Relative humidity (%)	30.72	70.51	78.88	77.80	86.11	100	10.99
Non-accidental	0	13.51	20.64	22.62	29.29	343.63	13.41
Gender							
Male	0	13.91	23.41	26.44	35.33	405.63	18.18
Female	0	8.14	15.77	18.61	25.64	297.33	14.78
Age							
<65	0	4.89	10.29	12.24	17.39	257.57	10.51
≥65	0	66.05	102.40	113.55	147.15	2428.45	72.20
Cardiovascular	0	5.00	8.82	10.14	13.65	225.50	7.48
Gender							
Male	0	4.19	9.35	11.41	16.12	255.20	10.21
Female	0	2.72	6.92	8.80	12.63	194.19	8.52
Age							
<65	0	0	2.66	4.05	6.49	122.79	5.34
≥65	0	31.09	54.96	63.63	85.63	1760.64	48.90
Subtypes							
Hypertension	0	0	1.89	2.93	4.43	100.22	3.57
Cerebrovascular	0	0	2.58	3.67	5.44	113.41	4.16
Hemorrhagic stroke	0	0	0	1.61	2.37	56.53	2.81
Ischemic heart stroke	0	0	0	0.83	1.10	100.16	1.72
Respiratory	0	0	1.35	2.33	3.38	65.45	3.23
Gender							
Male	0	0	0	2.80	4.12	105.64	4.67
Female	0	0	0	1.84	2.57	104.59	3.82
Age							
<65	0	0	0	0.76	0	58.36	2.47
≥65	0	0	9.49	15.75	24.40	488.11	20.80
Subtype							
COPD	0	0	0	0.58	0	24.15	1.39

SD: standard deviation; COPD: chronic obstructive pulmonary disease.; P_2.5_: 2.5th percentiles; P_97.5_: 97.5th percentiles.

**Table 2 ijerph-17-02699-t002:** The attributable risk of cold and heat ambient temperature on YLL in disease-specific populations.

Disease Death	Total (%)	Cold (%)	Heat (%)	Extreme Cold (%)	Moderate Cold (%)	Moderate Heat (%)	Extreme Heat (%)
Non-accidental	10.73 (4.36–17.09)	10.27 (4.52–16.03)	0.45 (−0.16–1.06)	1.02 (0.64–1.39)	9.26 (3.87–14.64)	0.22 (−0.23–0.68)	0.23 (0.08–0.38)
Gender							
Male	7.64 (0.71–14.57)	6.99 (1.21–12.76)	0.65 (−0.50–1.81)	0.83 (0.45–1.21)	6.16 (0.76–11.55)	0.44 (−0.50–1.38)	0.22 (0–0.43)
Female	14.99 (6.77–23.21)	14.59 (6.75–22.43)	0.39 (0.01–0.77)	1.20 (0.73–1.68)	13.39 (6.03–20.76)	0.14 (−0.09–0.36)	0.26 (0.10–0.41)
Age							
<65	5.23 (−2.47–12.92)	4.93 (−1.53–11.39)	0.30 (−0.94–1.54)	0.77 (0.34–1.19)	4.16 (−1.87–10.19)	0.20 (−0.79–1.18)	0.10 (−0.15–0.35)
≥65	14.92 (8.16–21.68)	14.06 (7.95–20.17)	0.86 (0.21–1.52)	1.19 (0.78–1.60)	12.87 (7.17–18.57)	0.50 (−0.03–1.02)	0.37 (0.24–0.50)
Cardiovascular	16.44 (9.09–23.79)	15.94 (8.82–23.05)	0.50 (0.26–0.73)	1.39 (0.99–1.79)	14.55 (7.83–21.27)	0.15 (0.05–0.26)	0.35 (0.22–0.48)
Gender							
Male	14.41 (4.59–24.23)	13.97 (4.44–23.5)	0.44 (0.15–0.73)	1.26 (0.78–1.75)	12.71 (3.67–21.76)	0.14 (0–0.27)	0.30 (0.15–0.45)
Female	17.90 (7.92–27.88)	17.35 (7.67–27.04)	0.54 (0.25–0.84)	1.47 (0.92–2.01)	15.89 (6.75–25.03)	0.17 (0.05–0.29)	0.37 (0.20–0.55)
Age							
<65	12.45 (−0.63–25.52)	12.16 (−0.56–24.89)	0.28 (−0.07–0.63)	1.00 (0.42–1.58)	11.16 (−0.99–23.31)	0.06 (−0.05–0.16)	0.23 (−0.02–0.48)
≥65	18.54 (10.93–26.15)	17.70 (10.73–24.67)	0.84 (0.20–1.47)	1.53 (1.06–2.00)	16.17 (9.66–22.67)	0.42 (−0.08–0.91)	0.42 (0.28–0.56)
Respiratory	5.47 (−2.65–13.60)	4.31 (−1.75–10.37)	1.16 (−0.91–3.22)	0.56 (0.11–1.01)	3.75 (−1.86–9.37)	0.86 (−0.85–2.56)	0.30 (−0.05–0.66)
Gender							
Male	5.41 (−4.20–15.01)	3.28 (−3.43–10.00)	2.12 (−0.76–5.01)	0.40 (−0.11–0.90)	2.88 (−3.33–9.10)	1.72 (−0.72–4.16)	0.40 (−0.04–0.85)
Female	7.48 (−11.65–26.61)	7.19 (−11.46–25.84)	0.29 (−0.19–0.77)	0.79 (−0.09–1.67)	6.40 (−11.37–24.17)	0.08 (−0.10–0.26)	0.21 (−0.09–0.52)
Age							
<65	15.40 (−7.68–38.48)	1.12 (0.34–1.89)	14.28 (−8.02–36.58)	1.01 (0.36–1.65)	0.11 (−0.02–0.24)	13.69 (−7.49–34.87)	0.59 (−0.53–1.72)
≥65	8.77 (0.39–17.15)	7.91 (1.16–14.65)	0.86 (−0.77–2.50)	0.68 (0.20–1.16)	7.23 (0.97–13.49)	0.55 (−0.76–1.87)	0.31 (−0.01–0.63)

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
