# Peer review of "The Impact of Non-optimum Ambient Temperature on Years of Life Lost: A Multi-county Observational Study in Hunan, China"

_ijerph, 2020, doi:10.3390/ijerph17082699_

Round 1

Reviewer 1 Report

The article presents an statistical approach based on a non lineal method to the study of the relationships between ambiente temperature and health on a specify area of China considering 

The novelty of the approach is important due to the used of years of life lost as the outcome of the impact of ambient temperature on health. This makes the study very interesting.

The aim of the study is clear and data sources and applied methods are well describe. Descriptions of parameters of the formulas should use a type of letter cursive or different to the normal text to facilitate reader the understanding of them.

Results are clearly presented and discussion includes update references such as the one to Coronavirus and temperature, Conclussions are perhaps  too short and a some more conclussions could be stated.

I was able to read properly the document in english and i consider it is fine but english is not my first spoken language. 

In any case ny valoration of the article is possitive and i consider is ready for publication.

Author Response

Thanks for your suggestion. We have revised the conclusions, based on helpful comment from you.

On line 32-33, we have revised the conclusions in the part of “Abstract”. As following:

This study highlights that general preventative measures could be important for moderate temperatures, whereas quick and effective measures should be provided for extreme temperatures

On line 296-301, we have revised the conclusions in the part of “Conclusions”. As following:

This study suggests that public health interventions to minimize the health consequences of non-optimum temperatures are necessary. General preventative measures, such as planting trees and installing air conditioners, should be done to minimize the health consequences of moderate temperatures. Whereas quick and effective measures should be provided for extreme temperatures, such as starting the emergency system and protecting outdoor workers.

Reviewer 2 Report

Overall, I find this article to be very fascinating but I would recommend a few changes before publication.  

First, the discussion does an excellent job of relating the findings with broader issues.  The conclusion, however, is too short and needs revision (or more accurately an addition).  First, summarize the key findings and how this is contributing, such as the cold temperature finding shifting some paradigms around the issue.  Second, offer some policy recommendations or ways to apply these findings.  What should be done with this new knowledge and is it translatable to other locations or is this a China-specific event?  Overall, the conclusion needs work before publication.  

Second, I would like to see a little more background and literature added into the introduction.  There is a lot of research on heat stress and yet, that is not discussed.  I think the finding around cold temperatures would be stronger and more impactful if the authors included more background on the literature and previous work done in "heat-related illness", particularly in US/Canada, Europe, and Australia. Again, the introduction should link to the revised conclusion to allow the authors to scale up the findings and make them more impactful globally.  

Reviewer 3 Report

Dear Authors,

Thank you for submitting your manuscript to this journal. Please address the following issues to improve the quality of this paper:

1- I don't understand the title. How can "ambient temperature" have impact on anything? It is about very high or very low ambient temperature which affects people.

2- Basically we can assume that there is no introduction/literature review in this paper! The authors have mentioned only a few previous works. Accordingly, the reader will not see the history and gaps in the work and the need for this work.

3- Per journal instruction, the references should be mentioned in "[]" and not in "()" and also in line 44-45, check the referencing method. Generally, the authors have to check the referencing method.

4- The hospitalization data covers only 5 years and in my opinion,  this is not an enough period of time for this study and conclusion.

5- Why the focus is on the ambient temperature, while we know that apparent temperature has more significant impact on health issues?

In addition, usually heatwave (and cold wave) which means a series of hot (or cold) days are very important in health issues and not only one day. I strongly recommend the authors to check the papers about the heatwaves (and how they are different from single day with high temperature) and also cold waves. Following is a few examples of papers, discussing about these issues:

I- Criteria for heat and cold wave duration indexes (https://link.springer.com/article/10.1007/s00704-011-0495-8)

II- Localized Changes in Heat Wave Properties Across the United States (https://doi.org/10.1029/2018EF001085)

III- Susceptibility to mortality related to temperature and heat and cold wave duration in the population of Stockholm County, Sweden (https://doi.org/10.3402/gha.v7.22737)

6- Temperature data and the way that they have been handled are questionable. For example, in urban area, UHI significantly increased health risks. This very important issue has been ignored.

7- YLL calculation is not clear at all. 

8- In sensitivity analysis, I didn't understand how and why it has been done. In addition, in the middle of a scientific paper, usually no one says: "The code is available if requested". 

9- What is the definition of "risk" in this work?

Thanks and Good Luck!

Round 2

Reviewer 3 Report

Thanks for addressing the mentioned issues!